# Workers’ Injury Risks Focusing on Body Parts in Reinforced Concrete Construction Projects

**DOI:** 10.3390/ijerph21121655

**Published:** 2024-12-11

**Authors:** Jiseon Lim, Jaehong Cho, Jeonghwan Kim, Sanghyeok Kang

**Affiliations:** 1Department of Civil & Environmental Engineering, Incheon National University, 119 Academy-ro, Yeonsu-gu, Incheon 22012, Republic of Korea; jslim1@samaneng.com (J.L.); jhong3787@inu.ac.kr (J.C.); 2Department of Civil Engineering, Korea National University of Transportation, 50 Daehak-ro, Chungju 27469, Republic of Korea; jeonghwan.kim@ut.ac.kr; 3Incheon Disaster Prevention Research Center, Incheon National University, 119 Academy-ro, Yeonsu-gu, Incheon 22012, Republic of Korea

**Keywords:** construction safety management, occupational accidents, worker’s injured body part, risk score, reinforced concrete construction

## Abstract

This study addresses occupational safety in reinforced concrete construction, an area marked by high accident rates and significant worker injury risks. By focusing on activity–body part (A–BP) combinations, this research introduces a novel framework for quantifying injury risks across construction activities. Reinforced concrete construction tasks are categorized into ten specific activities within three major work types: rebar work, formwork, and concrete placement. These are further analyzed concerning six critical body parts frequently injured on-site: head/face, arm/shoulder, wrist/hand, torso, leg/pelvis, and foot/ankle. Using data from 2283 construction accident reports and expert surveys, the probability and severity of injuries for each A–BP element were calculated. Probability scores were derived from actual incident data, while severity scores were determined via expert evaluations, considering injury impact and the required recovery time. To ensure precision and comparability, scores were standardized across scales, enabling a final risk assessment for each A–BP. Results identified that wrist and hand injuries during rebar work activities, particularly cutting and shaping, exhibited the highest risk, underscoring the need for focused protective measures. This study contributes to construction safety management by providing detailed insights into injury risk based on activity–body part interactions, offering safety managers data-driven recommendations for tailored protective equipment, enhanced training, and preventive protocols. This research framework not only helps optimize safety interventions on conventional construction sites but also establishes a basis for future studies aimed at adapting these strategies to evolving construction methods.

## 1. Introduction

The construction industry is known for its high frequency and severity of occupational accidents, making effective safety management essential. According to data from South Korea’s Ministry of Employment of Labor, 50.4% of all industrial accident fatalities occur in construction [1]. Particularly in South Korea, strict safety regulations such as the Serious Accidents Punishment Act have been enacted, allowing business suspensions for safety violations. As a result, safety management has become a critical factor not only for worker protection but also for the overall success of construction projects.

To address these challenges, various effects have been made to assess the risks associated with different construction activities [2,3,4,5]. For example, some studies have developed models to predict risk levels based on the type and frequency of work activities [2,3], while others focus on environmental factors that contribute to higher-risk scenarios [6,7]. Meanwhile, research on workers’ injured body parts has primarily analyzed correlations with factors such as occupation and age [8,9,10,11,12]. However, these studies often lack insights into which body parts are more vulnerable during specific construction activities.

While many studies attempt to assess general activity risks, there is a growing need for a more detailed understanding of specific risks. Activity-based risks and injury characteristics for each body part are often studied separately, resulting in limited discussions on how specific activities impact particular body parts and to what extent. Bridging this gap is crucial for enhancing injury prevention strategies, as it allows for the development of activity- and body-part-specific safety protocols, leading to a more comprehensive approach to safety management on construction sites. Therefore, this study proposes a new approach that calculates risk scores for risk elements linked to both activity and body parts. Specifically, we define activity–body parts as risk elements, rather than focusing solely on activity risk scores, and calculate risk scores for these combined elements.

This approach has the following expected benefits: Firstly, it allows us to assess the extent to which specific activities affect different body parts. Secondly, it provides concrete evidence for recommending and developing specialized protective equipment for each body part. Thirdly, it enhances worker awareness by providing information on vulnerable body parts for each activity during safety training.

The scope of this study focuses on reinforced concrete construction, a common operation across various projects, including foundation, retaining wall, tunnel, and road construction, and it constitutes a major portion of construction work. According to the 2021 statistics from Statistics Korea, reinforced concrete construction represents the largest share among the 14 specialized construction sectors, accounting for 14.18% of registered contractors [13]. Finally, reinforced concrete construction is labor-intensive, as many tasks are still performed manually due to limited mechanization and automation. This reliance on labor increases the risk of worker injuries.

This study begins by defining risk elements and developing a risk breakdown structure, categorizing injuries into six main body parts: (1) head/face, (2) arm/shoulder, (3) wrist/hand, (4) torso, (5) leg/pelvis, and (6) foot/ankle. Meanwhile, reinforced concrete construction was divided into three primary work types: (1) rebar work, (2) form and support work, and (3) concrete work. These categories were further broken down into specific activities: rebar work includes cutting and shaping rebar, transporting and lifting rebar, and placing rebar; form and support work consists of material preprocessing, transporting and lifting forms, positioning and alignment, and form removal and cleaning; finally, concrete work is divided into transporting and placing concrete, finishing and cleaning, and curing. Each of these ten activities was matched to the six body parts, resulting in sixty Activity–Body Part (A–BP) categories.

The risk score for each A–BP was quantitatively assessed by considering the probability of occurrence and the severity of injury. The probability was calculated based on actual accident data from Korea’s Ministry of Land, Infrastructure and Transport’s Construction Safety Management Integrated Information system (CSI) [14], while severity was determined through expert surveys. These scores were then combined to produce the final risk score. Figure 1 illustrates the procedure used in this study.

The structure of this study is as follows: Section 2 reviews the relevant literature, identifies the research gap, and details the methodology for evaluating risk scores for A–BPs. Section 3 presents the analysis results. Section 4 discusses the implications of the findings, and Section 5 provides a conclusion, outlining the study’s limitations and suggestions for further studies.

## 2. Materials and Methods

### 2.1. Workers’ Injury Risk in Construction Activity

Research on workers’ injury risks typically focuses on analyzing the hazards associated with construction activities or assessing the vulnerability of specific body parts within certain construction work types as shown in Table 1. Passmore et al. (2019) analyzed U.S. fatal accidents from 2015 to 2017 and found that head and neck injuries have the highest fatality risk, torso injuries have a moderate risk, and limb injuries such as those concerning fingers, hands, and wrists have the lowest fatality risk [15]. Jeong (1999) analyzed construction industrial accidents in South Korea and revealed that the body parts most vulnerable to injury in non-fatal accidents were the leg, foot, and toe, while the head, face, and neck were most vulnerable in fatal incidents [16]. Chi and Han (2013) found a significant correlation between the injury type and the part of the body injured in U.S. construction accidents; for example, fractures were commonly associated with injuries to the head and back [6]. Sugama and Ohnishi (2015) found that fractures are the most common injury from stepladder accidents in Japan, comprising 70% of injuries, with lower limbs (34.7%) and upper limbs (21.4%) being the most affected [17]. Choi (2015) revealed that, in the United States, the most frequently injured body part was the fingers/hands/wrists, accounting for 26% of injuries (i.e., 37 out of 143 cases), followed by the back (10%), foot/ankle (9%), eye (9%), multiple body parts (9%), and knee (8%) [8]. Amiri et al. (2014) found that hands and limbs are the most commonly injured body parts, with injuries to the cranium, brain, spine, back, and eyes resulting in the most severe consequences in Iran [18]. Similarly, Berglund et al. (2019) showed that in the Swedish construction industry, the most commonly injured body parts are the hand/wrist, fingers, leg (including the knee), and foot/ankle/toe [9]. Halvani et al. (2012) found a significant correlation between injured body parts and accident consequences, but no correlation between accident type and its consequences in Iran [10]. Mehrdad et al. (2014) indicated that falls from heights and crush injuries were the most prevalent types of accidents, with injuries most commonly occurring to the upper and lower extremities in Iran [7]. Fontaneda et al. (2022) found that Spanish construction workers under 39 mostly suffered extremity injuries, while those over 50 experienced more arm, shoulder, leg, and hip injuries [11].

Table 1 provides a systematic review of previous studies in construction safety research, categorizing them based on their analytical focus across three dimensions: Risk of Activity, Injury Risk of Workers’ Body Parts, and Risk of Activity–Body Part Combinations. This categorization reveals distinct patterns in research approaches. Studies by Casanovas et al. (2014) [3], Ayhan and Tokdemir (2020) [4], and Jannadi and Almishari (2003) [5] primarily focused on activity-based risk assessment, developing methodologies to evaluate and quantify risks associated with specific construction tasks. In contrast, researchers such as Jeong (1999) [16], Choi (2015) [8], and Berglund et al. (2019) [9] concentrated on analyzing injury patterns related to workers’ body parts, examining factors like injury frequency and severity. However, as indicated by the ‘X’ marks in the last column, none of these studies have attempted to integrate both perspectives by analyzing the specific relationships between construction activities and body part injuries.

The problem addressed in this study is that activity-based risks and injury characteristics for each body part are often studied separately, resulting in limited discussions on how specific activities impact particular body parts and to what extent. Previous studies have primarily focused on calculating risks associated with activities alone, with minimal research on the relationship between activities and injured body parts or on risks associated with their combined effects. Therefore, this study proposes a new approach that calculates risk scores for risk elements linked to both activity and body parts. Specifically, we define activity–body parts as risk elements, rather than focusing solely on activity risk scores, and calculate risk scores for these combined elements.

### 2.2. Risk Assessment Method

Risk assessment is defined as a methodology to identify and quantify potential hazards impacting project personnel, where severity and probability are the key components. Severity indicates the potential level of damage, while probability represents the likelihood of occurrence. Risk scores are typically calculated by combining these elements, often through multiplication or addition. In this study, we introduced a convert qualitative risk assessment method into quantitative metrics by evaluating both the probability and severity of A–BPs. Probability is classified into five levels with assigned numerical values: very high (5), high (4), moderate (3), low (2), and very low (1). Similarly, severity is divided into four levels: very severe injury (4), severe injury (3), moderate injury (2), and minor injury (1). This scoring approach facilitates a quantitative risk evaluation, allowing for the prioritization of mitigation strategies based on combined severity and probability scores. Further details on the classification process, including the importance of precise interval definitions for severity and probability, are discussed in the data section, emphasizing how these definitions help reduce bias and improve accuracy.

The Korea Occupational Safety and Health Agency (KOSHA) also suggests calculating risk scores by multiplying or adding severity and probability scores. For example, KOSHA classifies likelihoods as follows: very high = 5, high = 4, medium = 3, low = 2, very low = 1; while severity levels are categorized as death or disability (4), requiring time off work (3), not requiring time off work (2), non-treatment (1) [23]. The specific numerical evaluations, however, may vary based on the context in which they are applied. This score system allows flexibility in adapting risk assessment methodologies across different application areas, and is mathematically represented as follows:Risk Score = Severity (or Impact) Score * Probability (or Likelihood) Score

Many researchers have developed methods for quantitatively assessing risks in construction, generally following the concept of multiplying or adding severity and probability. Jannadi and Almishari (2003) proposed a model for calculating the Activity Risk Score in construction, based on the seriousness, exposure, and likelihood of potential hazards [5]. In their study, the Activity Risk Score is defined as follows:Activity Risk Score = Severity (S) × Exposure (E) × Probability (P)
where Severity (S) is the potential severity of the hazard associated with the activity, Exposure (E) represents the degree of exposure to the hazard, and Probability (P) is the likelihood that harm will occur if a hazardous event takes place. Lee et al. (2012) defined ‘site weight’ as the ratio of specific classification risk to average risk, considering work process rates and project costs [2]. Casanovas et al. (2014) defined the occupational risk index as the product of a standardized risk score and work time for hazardous activities, comparing risks in precast and in situ projects [3]. These methods highlight numerical evaluation efforts to identify high-risk construction activities.

The probability of A–BPs was determined using data from 2283 construction accident reports filed with Korea’s Ministry of Land, Infrastructure and Transport’s Construction Safety Management Integrated Information system (CSI) from January 2019 to December 2021 [14]. These reports included detailed information on accident dates, types of construction, causes, consequences, and incident descriptions. During the data preprocessing stage, additional categorization was needed since the initial data did not specify detailed activities or affected body parts. Therefore, keywords related to reinforced concrete construction (e.g., concrete pouring, formwork, shoring, and rebar) were used to create an activity field. Similarly, information on injured body parts was extracted from fields on causes, consequences, and incident descriptions to create an injured body part field. This classification allowed for an analysis of accident frequency across 60 A–BPs, representing ten specific activities and six body parts associated with reinforced concrete construction.

The severity of A–BPs is expected to vary across different scenarios. Due to the lack of detailed injury severity data from accident reports, an expert survey was conducted from 29 April to 5 May 2022, involving 110 experts from academia and the construction industry, with 53 responses collected. The survey was administered via email, asking experts to assign a severity score to each of 60 A–BPs, with the mean score set as the original severity score. Respondents included 63.5% from the civil and infrastructure sector, and 23.1% from the residential and building construction sector, with 15.4% academics and 82.7% professionals. Their experience varied, as follows: 23.1% had less than 5 years, 19.3% had 5–9 years, 21.2% had 10–14 years, and 36.5% had over 14 years of experience. The survey used injury severity levels from the workplace risk assessment guidelines provided by the Korean Ministry of Employment and Labor [24]. Injuries were categorized into four severity levels: (1) mild injuries with immediate return to work, (2) injuries requiring emergency treatment and time off, (3) serious injuries with extended absence, and (4) injuries leading to permanent disability or impairment.

#### Risk Breakdown Structure

This study focuses on reinforced concrete construction projects as its area of study. Based on a literature review and expert interviews, reinforced concrete construction projects were categorized into three work types (level 1): (1) rebar work, (2) form and support work, and (3) concrete work. These three categories were further divided into detailed activities, as is illustrated in Figure 2. Rebar work encompasses three activities: (1) cutting and shaping rebar, (2) transporting and lifting rebar, and (3) placing rebar. Form and support work is divided into four activities: (1) material preprocessing, (2) transporting and lifting forms, (3) positioning and alignment, and (4) form removal and cleaning. Concrete work, lastly, is broken down into three activities: (1) transporting and placing concrete, (2) finishing and cleaning, and (3) curing.

The emphasis of this study is on the injured body parts of workers. We defined the risk element as the activity–body part (A–BP) and aimed to calculate the risk score for each risk element. To achieve this, it is essential to classify human body parts. Terms for the injured body parts derived from descriptions of accident cases were various. For analysis uniformity, body parts were grouped into six categories: (1) H: head/face, (2) A: arm/shoulder, (3) W: wrist/hand, (4) T: torso, (5) L: leg/pelvis, and (6) F: foot/ankle. This categorization is tailored to the study’s goals, dividing the body into axial (head, neck, spine, sternum, scapula, and hip bones) and appendicular skeleton parts (upper and lower limbs). Categorizing the human body into specific parts is not a standardized process; rather, researchers tailor these classifications according to the unique goals of their studies [8,20]. Consequently, each activity was paired with these six body parts, resulting in a total of 60 A–BPs as shown in Figure 2.

### 2.3. Data

#### 2.3.1. Probability Scores of Activity–Body Parts

As outlined in the previous section, we organized 2283 cases of accidents that occurred in reinforced concrete construction projects in South Korea from January 2019 to December 2021, categorizing them by activity and the injured body part. Based on these data, we calculated the frequency of injuries to different body parts by activity. Table 2 illustrates the distribution of accident-prone body parts (A–BPs) across ten primary construction activities. Findings indicate that Form and Support work (F) accounted for the largest proportion of incidents, with 1635 cases (71.6%), followed by Reinforcement work (R) with 367 cases (16.1%), and Concrete work (C) with 281 cases (12.3%). Notably, the most accident-prone activity was Form positioning and alignment (F3), contributing to 807 cases (35.3%). Wrist and hand injuries (F3-W) were found to be the most common within this category, representing 30.5% of the accidents. Conversely, activities such as Concrete finishing and cleaning (C2), along with Concrete curing (C3), were the least prone to accidents, with 23 (1.0%) and 28 (1.2%) cases, respectively. The analysis also revealed that wrist and hand injuries were the most prevalent, occurring in 779 cases (34.1%), followed by foot and ankle injuries in 324 cases (14.3%). Arm and shoulder injuries were comparatively rare, with only 201 cases (8.8%).

Assigning probability scores to varying frequency intervals in risk analysis requires a careful approach, as the risk score is heavily influenced by classification criteria. The frequency distribution, as shown in Table 3, spans a broad range from 0 to 246, with significant variations. To address this, three methods were applied as follows: (1) dividing the range into equal intervals, (2) using a ranking-based method for interval division, and (3) utilizing scores based on a linguistic classification corresponding to annual frequency occurrences.

To ensure that the probability scores accurately represent the risk associated with each of the A–BPs, we compared three different methods for score calculation, as is detailed in Table 3. The first method involved dividing the frequency range into equal intervals. Although straightforward, this approach resulted in a skewed distribution, where the majority of A–BPs fell into the lowest score category. This imbalance did not accurately capture the variations in injury frequency and was therefore unsuitable for a nuanced risk analysis.

The second method used a ranking-based approach, which aimed to assign scores based on the relative frequency ranks of each A–BP. However, this method encountered practical difficulties due to overlapping frequency ranks, which made it challenging to assign distinct scores consistently. This overlap introduced ambiguity into the scoring system, potentially obscuring the true differences in risk levels between certain A–BPs.

The third method, which we ultimately selected, involved a linguistic classification aligned with occurrence intervals (e.g., “occurring less than once a year”, “occurring more than twice a month”) (Table 4). This approach provided a realistic and balanced distribution of probability scores by classifying A–BPs into categories that reflected actual annual and monthly occurrence patterns. This method allowed us to capture a more accurate representation of risk, especially for activities with widely varying frequencies, and facilitated a clearer, more interpretable scoring system for stakeholders.

The figures presented in Table 5 provide a detailed perspective on the probability of injury risks across various A–BPs in reinforced concrete construction. Probability scores indicate the likelihood of injuries occurring for specific A–BPs based on historical incident data. Higher scores correspond to A–BPs that have been observed more frequently in accident reports, signaling areas where workers face greater risks due to the nature of the task and the vulnerability of particular body parts. For instance, a high probability score for wrist–hand injuries during rebar cutting and shaping reflects a higher frequency of incidents in these activities, emphasizing the need for focused safety interventions, such as specialized personal protective equipment (PPE) or ergonomic tools to reduce strain. In contrast, lower probability scores for activities like concrete finishing and cleaning imply a lower observed frequency of injuries in those tasks, suggesting that while safety protocols are still important, these tasks may pose comparatively lower immediate risks to workers.

Table 5 presents the probability scores for different A–BPs in reinforced concrete construction, derived from the analysis of 2283 accident cases reported between 2019 and 2021. The probability scores range from 1 to 5, with higher scores indicating more frequent occurrences of accidents. Our analysis reveals notable patterns: form and support work shows the highest overall probability scores, particularly in positioning and alignment activities (F3) with scores predominantly at 4–5, indicating frequent accidents. Within specific body parts, wrist and hand injuries consistently demonstrate high probability scores (3–5) across multiple activities, especially in rebar cutting and shaping (R1) and form material preprocessing (F1). In contrast, concrete finishing (C2) and curing activities (C3) generally show lower probability scores (1–2), suggesting less frequent accidents.

#### 2.3.2. Severity Scores of Activity–Body Parts

Based on the survey response data, average severity scores for A–BPs were calculated as shown in Table 6. The highest average severity score was for form removal and cleaning (F4) at 2.59, while concrete finishing and cleaning (C2) had the lowest at 1.48. Among specific body parts, the head/face had the highest severity score at 2.34, and the torso had the lowest at 2.02. The highest individual severity score was observed in form removal and cleaning-head/face (F4-H) at 2.96, while the lowest was in concrete curing-torso (C3-T).

#### 2.3.3. Standardization of Probability and Severity Scores

The initial probability and severity scores of A–BPs were calculated as previously outlined. However, using these results directly can lead to issues due to scale differences. The probability score uses a 5-point scale (1 to 5), while the severity score uses a 4-point scale (1 to 4). This discrepancy could overemphasize probability scores. Additionally, the probability score ranges from 1 to 5 (range of 4), while the severity score ranges from 1.41 to 2.96 (range of 1.55). To standardize the scales, this study adjusted the maximum value of each group to 1.00, with proportional adjustments for other values. This standardization was applied to both probability and severity scores, with the modified scores shown in Table 7.

Normalization of Original Probability Scores (OPS) (Equation (1)):
(1)$$NPS=\frac{OPS-Mean\;OPS}{Standard\;deviation\;OPS}$$
where *NPS* represents the normalized probability score, derived from the OPS of A–BP, *OPS* stand for the original probability score of A–BP, *Mean OPS* indicates the average probability score within the OPS dataset, and *Standard deviation OPS* is the standard deviation of the probability scores of A–BPs.Adjustment of NPS to ensure the minimum value is adjusted to 1.00, preventing any values from being 0.00 or negative (Equation (2)):
(2)TPS=NPS−Min NPS+1.00
where *TPS* is a temporary probability score used to calculate the final Modified Probability Score, and *Min NPS* represents the lowest score within the NPS dataset.Division of TPS by the maximum value to standardize the group’s highest value at 1.00 (Equation (3)):
(3)$$MPS=\frac{TPS}{Max\;TPS}$$
where *MPS* denotes the Modified Probability Score, and *Max TPS* represents the highest value within the TPS dataset.

## 3. Results

The final risk scores of A–BPs were calculated by combining the modified probability score and the severity score, with both terms weighted equally at 50:50. By adjusting the ratio to 30:70, 40:60, 50:50, 60:40, and 70:30, we observed that variations in the ratio did not significantly affect the risk score. Therefore, we decided to maintain a 50:50 ratio. Table 8 presents the final risk scores for A–BPs, ranging from a maximum of 1.000 to a minimum of 0.230, with an average of 0.578 and a standard deviation of 0.195. The distribution showed a 1st quartile at 0.389, a median (2nd quartile) at 0.601, and a 3rd quartile at 0.726.

Notably, the highest risk score was observed for R1-W (cutting and shaping reber-wrist/hand), with a score of 1.000. This was followed by F3-H (form positioning and alignment-head/face) and F4-H (form removal and cleaning) with scores of 0.871 and 0.869, respectively. Conversely, C2-A (concrete finishing and cleaning-arm/shoulder) recorded the lowest score at 0.230, with C3-F (concrete curing-foot/ankle) and C2-H (concrete finishing and cleaning-head/face) also showing low scores of 0.254 and 0.268, respectively.

The average risk score for form and support work was the highest at 0.692, followed by rebar work at 0.585, and concrete work at 0.418, as shown in Table 9. Regarding injured body parts, the highest average risk score was observed for wrist/hand at 0.688, followed by head/face at 0.583, torso at 0.564, foot/ankle at 0.553, leg/pevis at 0.550, and arm/shoulder at 0.530. Within reinforced concrete construction, the wrist/hand was the most at-risk body part, with scores of 0.773 in rebar work, 0.804 in form and support work, and 0.447 in concrete work, being 32.1%, 16.1%, and 6.9% higher than the average for each type of work, respectively. The arm/shoulder showed lower injury risks, recording 0.540 in rebar work, 0.647 in form and support work, and 0.364 in concrete work, each 7.7%, 6.5%, and 13.0% below the average for each work type. Compared to the average injury risk for each work type, wrist/hand and head/face had higher injury risks in rebar work and form and support work, while in concrete work, the risks were higher for wrist/hand, torso, foot/ankle, and leg/pelvis.

Table 10 shows the distribution of risk scores for A–BPs classified into Low, Medium, High, and Very High based on quartiles. In form and support work, 15 out of 24 activities were Very High risk, and 4 were High risk. In concrete work, 12 out of 18 activities were at the Low risk level. In rebar work, 12 out of 18 activities were Medium risk. Within form and support work, F3-H (form positioning and alignment-head/face) had the highest risk score at 0.871, and F1-T (material preprocessing-torso) had the lowest at 0.331. In rebar work, R1-W (cutting and shaping rebar-wrist/hand) recorded the highest score at 1.000, and R1-T (cutting and shaping rebar-torso) the lowest at 0.453. In concrete work, C1-W (transporting and placing concrete-wrist/hand) had the highest score at 0.712, and C2-A (finishing and cleaning-arm/shoulder) the lowest at 0.230.

## 4. Discussion

The results of this study indicated that the average risk score for form and support work was the highest, at 0.692, with 62.5% of form and support work activities categorized as Very High risk. Formwork activities were generally assessed as having a higher risk of physical injury compared to rebar and concrete work. This elevated risk can be attributed to the complex physical demands, involving multiple body parts and intricate movements. Specifically, F1, F2, and F3 require repetitive hand and wrist movements, placing considerable strain on wrist flexor and extensor muscles, as well as increased risks of balance loss of being struck by heavy materials when handling large formwork panels. The high risk of musculoskeletal injuries is attributed to the substantial physical demands needed to position and align formwork, necessitating force across both upper and lower body parts, and requiring significant strength, which collectively contributes to the elevated risk rating.

In terms of injured body parts, the overall risk for wrist and hand injuries was high (0.688), with rebar work (R1-W) showing the highest risk. These results are consistent with Choi (2015), who analyzed 143 injury reports from heavy highway projects in the Midwestern United States between October 2004 and November 2006. Choi’s study found that the fingers, hand, and wrist were the most frequently injured body parts, accounting for 26% of all injuries (37 out of 143 cases) [8]. Similarly, Berglund et al. (2019) analyzed injury data reported to the Swedish Social Insurance Agency in 2016 and reported that fingers (1.03 injuries per 1000 workers) and hands/wrists (1.04 injuries per 1000 workers) were the most commonly injured body parts in the Swedish construction industry [9]. Additionally, Dethlefsen et al. (2022) examined 397 cases of construction-related injuries treated at the emergency department of Bern University Hospital from 2016 to 2020, finding that upper extremity injuries comprised 40.8% of all injuries among construction workers, with hand injuries being the most common, representing 23.2% of upper extremity injuries [12]. These findings align closely with the results of our study.

The rebar work process comprises three main stages: shaping the steel rebar to the required design dimensions using a rebar cutter, transporting the shaped rebar to the designated site, and finally placing it at the specified intervals. Among all assessed risk factors, R1-W had the highest injury risk score and the most significant variance from the mean score, indicating the increased likelihood and severity of wrist and hand injuries during rebar cutting and handling activities.

Rebar processing often involves motorized or hydraulic rebar bending machines, which present specific risks to the hands, including glove entrapment and musculoskeletal disorders due to repetitive wrist exertion. The repetitive contact between workers’ hands and the machine’s hydraulic actuators, coupled with the high frequency of hand use, likely contributes to the elevated injury risks associated with rebar work. Additionally, arms and shoulders were rated with the second-highest risk in R1, primarily due to the use of conventional bending tools on off-grid sites, which rely on manual strength and repetitive motion, thereby increasing the likelihood of muscle fatigue and overuse injuries from cutting activities.

Among rebar activities, R2 recorded the highest risk score, particularly due to the hazards associated with rebar transport. Head, face, arms, and shoulders were rated as high-risk areas during this work. Typically, cranes or forklifts are used to transport and install rebar; while rare, incidents involving dropped loads can cause severe injuries, with the impact most likely concentrated on the head. Such injuries are severe due to the potential for concussions and other head trauma. Transporting long rebar, sometimes reaching up to ten meters, often requires ironworkers to carry rebar bundles on their shoulders, which poses a substantial strain on the shoulder and arm muscles, increasing the risk of musculoskeletal disorders. Additionally, the rebar’s small diameter, especially in a cross-sectional view, makes it difficult to detect visually, increasing the chance of head or face impacts on the job site. This limited visibility is exacerbated by the congested environment where workers must navigate through a densely arranged rebar, further raising the risk of sprains or lower back injuries if a worker’s foot becomes caught. Consequently, R2 poses a significantly higher risk level than other activities.

R3 involves spacing and tying rebar, which is generally associated with a lower overall risk level compared to other activities. However, the repetitive use of the wrists and hands during rebar tying increases the potential for chronic conditions such as carpal tunnel syndrome, indicating a relatively higher risk for these body parts. Recent advances in ergonomic power tools have shown promise in reducing injury risks associated with repetitive tasks, with major power tool manufacturers now offering various automated devices to support these functions. In light of the skilled labor shortage and the increasing importance of construction productivity, these technologies are gaining significant attention. As the adoption of such tools becomes more widespread, it is expected that physical injury risks will continue to decline.

The body-part-specific injury risk scores proposed in this study are based on a quantitative assessment system, revealing distinct injury risk levels associated with rebar, form and support, and concrete work. These findings can be directly applied in decision-making processes for safety managers, particularly in refining regulations and guidelines for PPE. Currently, OSHA classifies PPE guidelines into broad categories, including general construction, elevated work, electrical work, welding and cutting, and chemical handling. Each category recommends specific PPE, such as fall protection, insulated gloves, welding gear, or respiratory protection. Integrating the findings of this study into safety guidelines could facilitate more precise instructions on PPE requirements and work procedures tailored to each construction activity. For instance, rebar work could mandate clear polycarbonate face shields, safety goggles, or specialized shoulder pads to protect workers’ shoulders from rebar bundles. Wearable exoskeletons may also be recommended to minimize physical strain on ironworkers. Additionally, the study’s insights can inform the selection and quantity of first-aid supplies on-site based on the types of potential injuries.

Beyond PPE requirements, this study offers guidance for implementing safer work methods and design modifications. For example, spacer rebar could be redesigned into more stable configurations or installed in greater quantities to enhance leg support stability, thereby reducing physical strain on workers. Similarly, tasks involving complex rebar configurations, like shear rebar installation, could benefit from modular design approaches to simplify installation and minimize repetitive motions. Improving the layout of rebar storage and workspace areas to minimize unnecessary movement, as well as leveraging cranes and automated technologies to reduce manual rebar handling, are other practical recommendations arising from this study.

## 5. Conclusions

This study presents a structured approach to assessing injury risks by analyzing specific A–BP combinations within reinforced concrete construction. Through the analysis of 2283 accident reports and expert evaluations, we quantified injury probability and severity for each construction activity. Focusing on distinct activities, this study identifies high-risk A–BPs, particularly highlighting elevated injury risks to wrists and hands during rebar cutting and shaping tasks. These findings are crucial for safety management, providing a detailed risk map for injury-prone activities and enabling effective prioritization of safety interventions. The methodological framework emphasizes the importance of moving beyond generalized risk assessments toward a more nuanced understanding of injury risks in reinforced concrete construction.

Our A–BP classification method emphasizes activity-specific risk elements to enhance construction safety management practices. The findings demonstrate varying risk levels associated with specific activities, supporting the development of targeted protective strategies. This systematic categorization provides comprehensive insights into how construction activities contribute to injury risks, enabling a precise risk assessment. The methodology also establishes a foundation for investigating correlations between activity types and injury patterns, facilitating adaptive safety protocols across different construction scenarios.

The practical implications are significant for construction safety management. By providing quantitative A–BP injury risk data, this research enables safety managers to identify vulnerable body parts during specific construction activities. These insights support the customization of safety equipment, worker training programs, and activity-specific safety protocols. For instance, understanding the high-risk nature of wrist and hand injuries in rebar-related tasks can guide the selection of appropriate protective equipment and ergonomic tools. This data-driven approach facilitates proactive decision-making in resource allocation and workplace safety enhancement.

The study acknowledges limitations including the potential bias in accident reports due to incomplete or subjective information, and the exclusion of fatal accidents due to data availability constraints. Future research opportunities include expanding the study to other construction types and investigating additional risk factors such as environmental conditions, seasonal effects, and work duration variations. Such expanded analysis could provide more comprehensive insights into construction injury risks, enhancing safety management strategies across diverse construction environments.

## Figures and Tables

**Figure 1 ijerph-21-01655-f001:**
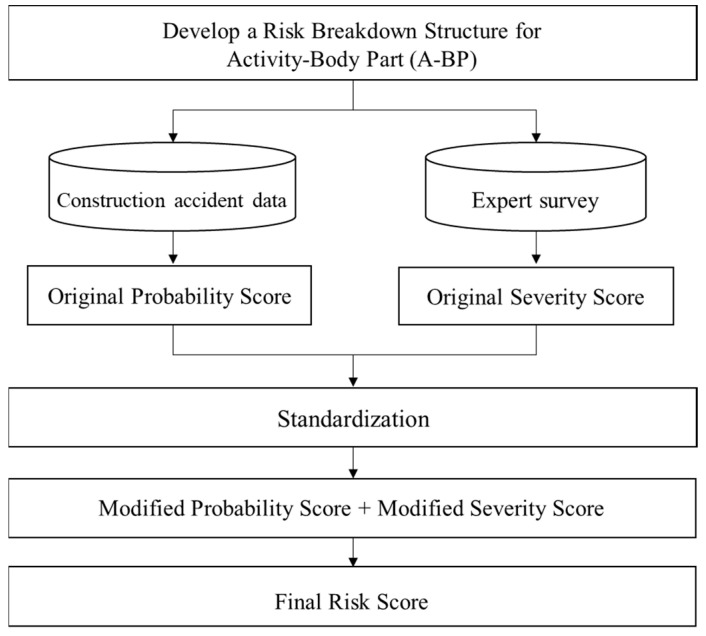
Research procedure.

**Figure 2 ijerph-21-01655-f002:**
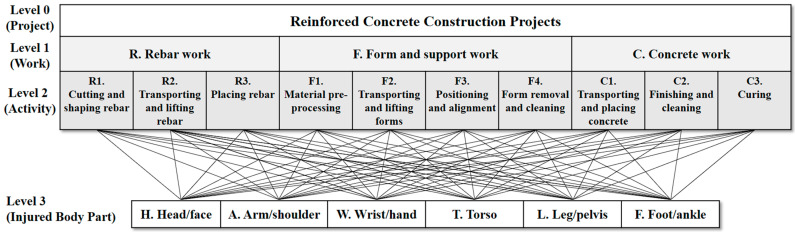
Risk element definition: activity–body part in reinforced concrete construction.

**Table 1 ijerph-21-01655-t001:** Summary of previous studies.

Authors	Risk of Activity	Injury Risk of Workers’ Body Parts	Risk of Activity–Body Part Combinations
Lee et al. (2012) [2]	X	X	X
Casanovas et al. (2014) [3]	O	X	X
Ayhan and Tokdemir (2020) [4]	O	X	X
Jannadi and Almishari (2003) [5]	O	X	X
Chi and Han (2013) [6]	X	O	X
Mehrdad et al. (2014) [7]	X	O	X
Choi (2015) [8]	X	O	X
Berglund et al. (2019) [9]	X	O	X
Halvani et al. (2012) [10]	X	O	X
Fontaneda et al. (2022) [11]	X	O	X
Dethlefsen et al. (2022) [12]	X	O	X
Passmore et al. (2019) [15]	X	X	X
Jeong (1999) [16]	X	O	X
Sugama and Ohnishi (2015) [17]	X	X	X
Ivaz et al. (2021) [19]	X	O	X
Yi (2018) [20]	X	X	X
Mistikoglu et al. (2015) [21]	O	X	X
Hallowell and Gabatese (2009) [22]	O	X	X

**Table 2 ijerph-21-01655-t002:** Frequency of activity–body parts in reinforced concrete construction for three years.

Work	Activity	H. Head/Face	A. Arm/Shoulder	W. Wrist/Hand	T. Torso	L. Leg/Pelvis	F. Foot/Ankle	Total
R. Rebar work		41 (11.2%)	25 (6.8%)	148 (40.3%)	58 (15.8%)	58 (15.8%)	37 (10.1%)	367 (100.0%)
	R1. Cutting and shaping rebar	8 (7.6%)	1 (1.0%)	83 (79.0%)	4 (3.8%)	5 (4.8%)	4 (3.8%)	105 (100.0%)
	R2. Transporting and lifting rebar	6 (10.2%)	7 (11.9%)	19 (32.2%)	13 (22.0%)	7 (11.9%)	7 (11.9%)	59 (100.0%)
	R3. Placing rebar	27 (13.3%)	17 (8.4%)	46 (22.7%)	41 (20.2%)	46 (22.7%)	26 (12.8%)	203 (100.0%)
F. Form and support work		190 (11.6%)	151 (9.2%)	567 (34.7%)	286 (17.5%)	202 (12.4%)	239 (14.6%)	1635 (100.0%)
	F1 Material preprocessing	3 (3.3%)	3 (3.3%)	78 (84.8%)	2 (2.2%)	4 (4.3%)	2 (2.2%)	92 (100.0%)
	F2. Transporting and lifting forms	14 (6.7%)	20 (9.6%)	72 (34.6%)	32 (15.4%)	31 (14.9%)	39 (18.8%)	208 (100.0%)
	F3. Positioning and alignment	104 (12.9%)	69 (8.6%)	246 (30.5%)	163 (20.2%)	102 (12.6%)	123 (15.2%)	807 (100.0%)
	F4. Form removal and cleaning	69 (13.1%)	59 (11.2%)	171 (32.4%)	89 (16.9%)	65 (12.3%)	75 (14.2%)	528 (100.0%)
C. Concrete work		30 (10.7%)	25 (8.9%)	64 (22.8%)	68 (24.2%)	46 (16.4%)	48 (17.1%)	281 (100.0%)
	C1. Transporting and placing concrete	27 (11.7%)	22 (9.6%)	49 (21.3%)	54 (23.5%)	36 (15.7%)	42 (18.3%)	230 (100.0%)
	C2. Finishing and cleaning	2 (8.7%)	3 (13.0%)	3 (13.0%)	8 (34.8%)	6 (26.1%)	1 (4.3%)	23 (100.0%)
	C3. Curing	1 (3.6%)	0 (0.0%)	12 (42.9%)	6 (21.4%)	4 (14.3%)	5 (17.9%)	28 (100.0%)
Total		261 (11.4%)	201 (8.8%)	779 (34.1%)	412 (18.0%)	306 (13.4%)	324 (14.2%)	2283 (100.0%)

**Table 3 ijerph-21-01655-t003:** Frequency distribution of categorization methods.

Classification	Categories	Frequency	Ratio (%)
Method 1. Equal Intervals	0~49	44	73.3
	50~99	10	16.7
	100~149	3	5.0
	150~199	2	3.3
	Over 200	1	1.7
Method 2. Ranking-Based	1~12	13	21.7
	13~24	11	18.3
	25~36	13	21.7
	37~48	12	20.0
	49~60	11	18.3
Method 3. Linguistic Classification	Occurring less than once a year	11	18.3
	Occurring less than once per quarter	15	25.0
	Occurring less than once a month	12	20.0
	Occurring more than once a month but less than twice a month	12	20.0
	Occurring more than twice a month	10	16.7

**Table 4 ijerph-21-01655-t004:** Frequency classification criteria for original probability scores.

Linguistic Classification	Probability Score	Description	Yearly Occurrence
Improbable	1	Occurring less than once a year	0~1
Remote	2	Occurring less than once per quarter	2~4
Occasional	3	Occurring less than once a month	5~12
Probable	4	Occurring more than once a month but less than twice a month	13~24
Frequent	5	Occurring more than twice a month	Over 24

**Table 5 ijerph-21-01655-t005:** Original probability scores of activity–body parts.

Activity	1. H: Head/Face	2. A: Arm/Shoulder	3. W: Wrist/Hand	4. T: Torso	5. L: Leg/Pelvis	6. F: Foot/Ankle	Mean
1. R1: Cutting and shaping reber	2	1	5	2	2	2	2.33
2. R2: Transporting and lifting reber	2	2	3	3	2	2	2.33
3. R3: Placing reber	3	3	4	4	4	3	3.50
4. F1: Material preprocessing	1	1	5	1	2	1	1.83
5. F2: Transporting and lifting forms	3	3	4	3	3	4	3.33
6. F3: Positioning and alignment	5	4	5	5	5	5	4.83
7. F4: Form removal and cleaning	4	4	5	5	4	5	4.50
8. C1: Transporting and placing concrete	3	3	4	4	3	4	3.50
9. C2: Finishing and cleaning	1	1	2	2	2	2	1.67
10. C3: Curing	1	1	1	2	2	1	1.33
Mean	2.50	2.30	3.80	3.10	2.90	2.90	2.92

**Table 6 ijerph-21-01655-t006:** Original severity scores of activity–body parts.

Activity	H. Head/Face	A. Arm/Shoulder	W. Wrist/Hand	T. Torso	L. Leg/Pelvis	F. Foot/Ankle	Mean
R1. Cutting and shaping reber	2.45	2.53	3.12	1.98	2.04	2.06	2.36
R2. Transporting and lifting reber	2.92	2.63	2.51	2.45	2.35	2.39	2.54
R3. Placing reber	1.98	1.90	2.02	1.67	1.80	1.84	1.87
F1 Material preprocessing	2.08	2.18	2.31	1.86	1.86	1.92	2.04
F2. Transporting and lifting forms	2.73	2.51	2.43	2.45	2.31	2.35	2.46
F3. Positioning and alignment	2.57	2.35	2.31	2.24	2.31	2.25	2.34
F4. Form removal and cleaning	2.96	2.61	2.49	2.53	2.49	2.45	2.59
C1. Transporting and placing concrete	2.33	2.18	2.29	2.12	2.24	2.24	2.23
C2. Finishing and cleaning	1.59	1.43	1.43	1.47	1.49	1.49	1.48
C3. Curing	1.75	1.59	1.75	1.41	1.47	1.53	1.58
Mean	2.34	2.19	2.27	2.02	2.04	2.05	2.15

**Table 7 ijerph-21-01655-t007:** Modified probability/severity scores of activity–body parts.

Activity	H. Head/Face	A. Arm/Shoulder	W. Wrist/Hand	T. Torso	L. Leg/Pelvis	F. Foot/Ankle
R1. Cutting and shaping reber	0.441/0.685	0.255/0.723	1.000/1.000	0.441/0.464	0.441/0.492	0.441/0.502
R2. Transporting and lifting reber	0.441/0.906	0.441/0.770	0.627/0.713	0.627/0.685	0.441/0.638	0.441/0.657
R3. Placing reber	0.627/0.464	0.627/0.427	0.814/0.483	0.814/0.318	0.814/0.380	0.627/0.398
F1 Material preprocessing	0.255/0.511	0.255/0.558	1.000/0.619	0.255/0.408	0.441/0.408	0.255/0.439
F2. Transporting and lifting forms	0.627/0.817	0.627/0.713	0.814/0.676	0.627/0.685	0.627/0.619	0.814/0.638
F3. Positioning and alignment	1.000/0.741	0.814/0.638	1.000/0.619	1.000/0.586	1.000/0.619	1.000/0.591
F4. Form removal and cleaning	0.814/0.925	0.814/0.760	1.000/0.704	1.000/0.723	0.814/0.704	1.000/0.685
C1. Transporting and placing concrete	0.627/0.629	0.627/0.558	0.814/0.610	0.814/0.530	0.627/0.586	0.814/0.586
C2. Finishing and cleaning	0.255/0.281	0.255/0.206	0.441/0.206	0.441/0.224	0.441/0.234	0.441/0.234
C3. Curing	0.255/0.356	0.255/0.281	0.255/0.356	0.441/0.196	0.441/0.224	0.255/0.253

**Table 8 ijerph-21-01655-t008:** Risk scores of activity–body parts.

Activity	H. Head/Face	A. Arm/Shoulder	W. Wrist/Hand	T. Torso	L. Leg/Pelvis	F. Foot/Ankle
R1. Cutting and shaping reber	0.563	0.489	1.000	0.453	0.467	0.471
R2. Transporting and lifting reber	0.674	0.605	0.670	0.656	0.540	0.549
R3. Placing reber	0.546	0.527	0.648	0.566	0.597	0.513
F1 Material preprocessing	0.383	0.407	0.810	0.331	0.424	0.345
F2. Transporting and lifting forms	0.722	0.670	0.745	0.656	0.623	0.726
F3. Positioning and alignment	0.871	0.726	0.810	0.793	0.810	0.796
F4. Form removal and cleaning	0.869	0.787	0.852	0.861	0.759	0.843
C1. Transporting and placing concrete	0.628	0.593	0.712	0.672	0.607	0.700
C2. Finishing and cleaning	0.268	0.230	0.323	0.333	0.338	0.338
C3. Curing	0.305	0.268	0.305	0.319	0.333	0.254

**Table 9 ijerph-21-01655-t009:** Average risk scores by work–body parts.

Work	H. Head/Face	A. Arm/Shoulder	W. Wrist/Hand	T. Torso	L. Leg/Pelvis	F. Foot/Ankle	Mean
R. Rebar work	0.594	0.540	0.773	0.558	0.534	0.511	0.585
F. Form and support work	0.711	0.647	0.804	0.661	0.654	0.677	0.692
C. Concrete work	0.400	0.364	0.447	0.441	0.426	0.430	0.418
Mean	0.583	0.530	0.688	0.564	0.550	0.553	0.578

**Table 10 ijerph-21-01655-t010:** Distribution of risk scores of activity–body parts.

Range	Risk Level	Rebar Work	Form and Support Work	Concrete Work	Total
Min–1st Q. (0.230–0.389)	Low	0 (0.0%)	3 (12.5%)	12 (66.7%)	15 (25.0%)
1st Q.–2nd Q. (0.389–0.601)	Medium	12 (66.7%)	2 (8.3%)	1 (5.6%)	15 (25.0%)
2nd Q.–3rd Q. (0.601–0.726)	High	5 (27.8%)	4 (16.7%)	5 (27.8%)	14 (23.3%)
3rd Q.–Ma x(0.726–1.000)	Very High	1 (5.6%)	15 (62.5%)	0 (0.0%)	16 (26.7%)
Total		18	24	18	60

## Data Availability

The datasets used in this study are available from the Construction Safety Management Integrated Information, http://www.csi.go.kr/acd/acdCaseList.do, accessed on 29 April 2022 (In Korean).

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
