# Peer review of "Workers’ Injury Risks Focusing on Body Parts in Reinforced Concrete Construction Projects"

_ijerph, 2024, doi:10.3390/ijerph21121655_

Round 1
Reviewer 1 Report
Comments and Suggestions for Authors
1. The problem statement or the research gap of the paper is lacking.
2. What were the previous study done on this? What were the results of the previous study/research? Why need to improve or need another study?
3. Why body parts and why reinforced concrete construction?
4. What is the significant of the body parts and reinforced concrete construction to the whole lifecycle of the project, or the performance of the project?
5. What has been done previously to improve those risks?
6. The study seems to focused on the risks to injuries and cause of the injuries in isolation from one activity to another activities.
7. Discussion and analysis need to be improved on relevant measures needed to mitigate the risks.
Thank you
Author Response
Comments 1: The problem statement or the research gap of the paper is lacking.
→ The problem addressed in this study is that activity-based risks and injury characteristics for each body part are often studied separately, resulting in limited discussion on how specific activities impact particular body parts and to what extent. Previous studies have primarily focused on calculating risks associated with activities alone, with minimal research on the relationship between activities and injured body parts or on risks associated with their combined effects. This issue has been clearly articulated in the introduction. Please refer to the second and third paragraphs of the Introduction.
To address these challenges, various effects have been made to assess the risks associated with different construction activities (Lee et al., 2012; Casanovas et al., 2014; Ayhan and Tokdemir, 2020, Jannadi and Almishari, 2003). For example, some studies have developed models to predict risk levels based on the type and frequency of work activities, while others focus on environmental factors that contribute to higher-risk scenarios. Meanwhile, research on workers’ injured body parts has primarily analyzed correlations with factors such as occupation and age (Choi, 2015; Berglund et al., 2019; Halvani et al., 2012; Chi and Han, 2013; Fontaneda et al., 2022; Dethlefsen et al., 2022). However, these studies often lack insights into which body parts are more vulnerable during specific construction activities.
While many studies attempt to assess general activity risks, there is a growing need for a more detailed understanding of specific risks. Activity-based risks and injury characteristics for each body part are often studied separately. Resulting in limited discussion on how specific activities impact particular body parts and to what extent. Bridging this gap is crucial for enhancing injury prevention strategies, as it allows for the development of activity- and body-part-specific safety protocols, leading to a more comprehensive approach to safety management on construction sites. Therefore, this study proposes a new approach that calculates risk scores for risk elements linked to both activity and body parts. Specifically, we define activity-body parts as risk elements, rather than focusing solely on activity risk scores, and calculate risk scores for these combined elements.
Comments 2: What were the previous study done on this? What were the results of the previous study/research? Why need to improve or need another study?
→ We have included Table 1 in the revised manuscript to provide a comprehensive summary of existing research on workers’ injury risks and to clearly illustrate the gap between previous studies and our research. Studies on workers’ injury risks typically focused on analyzing the hazards associated with construction activities or assessing the vulnerability of specific body parts within certain construction work. However, these studies often treat activity-based risks and injury characteristics for each body part independently, which limits understanding of how specific construction activities impact certain body parts and to what extent.
In particular, prior research has mainly concentrated on quantifying risks related to individual activities, with limited exploration into the interplay between activities and the specific body parts most affected during these activities. Consequently, there is minimal research addressing the combined effects of activities and body-part injuries, leaving a significant gap in how risk is comprehensively understood and managed in construction settings. Our study addresses this gap by introducing a new approach that calculates risk scores for "activity-body part" (A-BP) combinations. Instead of isolating activity risk scores, we define these A-BPs as distinct risk elements and calculate risk scores accordingly.
This study specifically addresses the need for improvement in construction safety management by focusing on risks associated with A-BPs within reinforced concrete construction—a sector characterized by high injury rates and labor-intensive work. While prior studies offer a general assessment of risks across construction activities, few have delved into the specific connection between individual tasks and the body parts most susceptible to injury during these activities. By identifying these A-BP-specific risks, our research contributes essential insights that support more tailored and effective safety interventions. This is particularly relevant for reinforced concrete construction, where general safety protocols often fail to capture the detailed risks associated with each activity.
To underscore the importance of our study, we have revised the manuscript to highlight how our framework enables safety managers to develop targeted interventions for each A-BP combination. This specificity enhances the precision and effectiveness of safety measures by addressing unique risks at the task level. Our approach also facilitates data-driven recommendations that are practical and directly applicable, such as selecting activity-specific personal protective equipment (PPE) and designing focused training that addresses the identified risks in a meaningful way.
In summary, our study not only addresses an underexplored area by analyzing A-BP-specific risks but also provides a valuable foundation for enhancing construction safety practices. By bridging the gap between general activity risks and specific injury risks, our findings offer a more precise and structured approach to construction safety management, which could improve safety protocols across both traditional and modern construction methods. We appreciate the reviewer’s insights, which have guided us in enhancing the clarity and impact of our manuscript to better communicate these significant contributions.
Comments 3: Why body parts and why reinforced concrete construction?
→ The reason this study focuses on workers' body parts is that, while many studies attempt to assess general activity risks, there is a growing need for a more detailed understanding of specific risks. Activity-based risks and injury characteristics for each body part are often studied separately, resulting in limited discussion on how specific activities impact particular body parts and to what extent. Bridging this gap is crucial for enhancing injury prevention strategies, as it enables the development of activity- and body-part-specific safety protocols, leading to a more comprehensive approach to safety management on construction sites. Therefore, this study proposes a new approach that calculates risk scores for risk elements linked to both activity and body parts. Specifically, we define activity-body parts as risk elements, rather than focusing solely on activity risk scores, and calculate risk scores for these combined elements. This rationale has been included in the introduction.
The study scope was set to reinforced concrete construction because it is a common operation across various projects, including foundation, retaining wall, tunnel, and road construction, and it constitutes a major portion of construction work. According to the 2021 statistics from Statistics Korea, reinforced concrete construction represents the largest share among the 14 specialized construction sectors, accounting for 14.18% of registered contractors. Finally, reinforced concrete construction remains labor-intensive, as many tasks are still performed manually due to limited mechanization and automation, which increases the risk of worker injuries. This information has been included in the introduction.
Comments 4: What is the significant of the body parts and reinforced concrete construction to the whole lifecycle of the project, or the performance of the project?
→ Focusing on body parts and reinforced concrete construction is especially significant during the construction stage of a project’s lifecycle, where detailed safety management can yield substantial benefits. This study’s identification of high-risk A-BPs aids in pre-construction planning by informing optimal work area layouts and construction scheduling, thereby minimizing exposure to hazards. This data also serves as a foundation for introducing tailored safety equipment and tools, aligning resources with specific risks.
In the construction stage, body-part-specific safety information supports a systematic approach to managing on-site safety. Detailed insights into vulnerable body parts per activity enable more effective training programs and safety briefings, heightening workers' awareness of particular risks. Additionally, the study’s findings facilitate the integration of customized safety equipment and PPE suited to specific tasks, and serve as a reference for sequencing and adapting work methods to further enhance safety.
Such a structured approach ensures a safer working environment by improving safety awareness, securing the workspace, and actively preventing accidents. Prioritizing targeted risk management in the construction stage not only safeguards workers but also contributes to project efficiency and performance, establishing a foundation for improved occupational health practices in the industry.
We have included the significance of the body parts and reinforced concrete construction to the performance of the project.
Comments 5: What has been done previously to improve those risks?
→ In previous research, various strategies have been implemented to mitigate injury risks in the construction industry, particularly focusing on general activity-based hazards and protective measures. Studies have examined standard risk factors associated with construction activities, such as the use of heavy machinery, work at elevated heights, and repetitive tasks, which are all linked to high injury rates. To address these risks, broad safety protocols and regulations, including the use of protective equipment like helmets, gloves, and harnesses, have been established as foundational measures in construction safety. These general interventions are designed to reduce common injury risks across construction sites, though they are not tailored to the specific characteristics of each construction task or the particular body parts most vulnerable during different activities.
Several studies have also attempted to refine construction safety by introducing more specific safety frameworks, including task-based risk assessments and ergonomic interventions. For example, certain research efforts have focused on identifying ergonomic hazards in repetitive tasks, aiming to reduce strain injuries by recommending ergonomic tools or equipment adjustments. Other studies have developed models to predict risk levels by factoring in both the frequency of task performance and environmental conditions. These task-based approaches have enhanced our understanding of the risks associated with specific construction activities but often lack detailed insights into how these tasks affect specific body parts, such as the hands, shoulders, or back, in labor-intensive activities like rebar work.
To further improve construction safety, recent research has proposed the use of data-driven safety management tools, such as wearable sensors and real-time monitoring systems, to identify high-risk behaviors and respond dynamically to evolving site conditions. These technologies can capture real-time data on worker posture, movements, and proximity to hazards, providing a detailed picture of risk in various tasks. However, while these tools represent a significant advancement in general safety monitoring, they are often not configured to analyze the relationship between specific activities and body part injuries. As a result, they may not offer the precise insights necessary for managing risks associated with particular body parts affected by specific construction tasks, such as wrist injuries in rebar shaping or shoulder strain in lifting tasks.
Our study builds upon these prior efforts by introducing a novel approach that focuses specifically on the injury risks associated with activity-body part (A-BP) combinations in reinforced concrete construction. By calculating risk scores for each A-BP element, this study bridges the gap between general task-based assessments and body-part-specific injury risks, providing a more nuanced perspective on safety management. This targeted approach allows for more precise interventions, such as selecting tailored personal protective equipment (PPE) and developing focused training programs that address the specific risks inherent in each activity. We believe this approach advances the existing body of knowledge by integrating activity and body-part risks, which could serve as a foundation for more effective safety protocols in both conventional and emerging construction practices.
Comments 6: The study seems to focused on the risks to injuries and cause of the injuries in isolation from one activity to another activities.
→ Thank you for your insightful comment. This study is based on the premise that each activity within reinforced concrete construction is conducted independently by separate crews, with minimal interaction between activities. Each activity - such as rebar work, form and support work, and concrete work - has distinct tasks that are not influenced by other activities. Therefore, we approached the analysis with the assumption that the risks and causes of injuries within each activity do not affect, nor are affected by, the risks in other activities.
This assumption aligns with the real-world structure of reinforced concrete construction projects, where each activity is typically carried out in isolation. By focusing on the unique risks inherent to each activity, the study aims to provide clear and tailored safety recommendations for each crew. Analyzing activities in isolation allows for a more practical and specific approach to safety management that matches the operational reality on construction sites.
Comments 7: Discussion and analysis need to be improved on relevant measures needed to mitigate the risks.
→ Thank you for your constructive comments. We appreciate your suggestion to delve deeper into the linkage between injured body parts and the characteristics of each activity. In response, we have incorporated detailed analyses in the discussion. Please refer to the first five paragraphs of the discussion section. The revised discussion sections now highlight the connections between specific activities and the affected body parts to better explain the nature of these injuries.
Reviewer 2 Report
Comments and Suggestions for Authors
The topic is exciting and practical. From the point of view of this reviewer, the manuscript could be accepted after addressing the following questions.
i). Data in Table 1 needs more clarification. How suddenly did data come?
ii). Tables 2 and 3: How is the score authentic per the Author’s opinion? The authors should give the meaning of the data.
iii). The introduction has to be more precise and focused. The topics to be covered in subsequent chapters should be clear.
v). The manuscript lacks its depth and originality.
vi). Conclusion needs to be thoroughly improved or changed.
vii). The work is not so sound in the academic study.
viii). The study was based on the conventional construction method. What about the modern construction approach?
ix). The following topic or its broader area may help you to improve your study.
# Safety barrier warning system for underground construction sites using Internet-of-Things technologies.
# Third-party quality control concept, methodology, and implementation in the building and housing construction industry in India
# Accident analysis for construction safety using latent class clustering and artificial neural networks.
# A comparative study of project risk management with risk breakdown structure (RBS): a case of commercial construction in India.
The manuscript needs thorough revision. Novelty is very low. The conclusion is not sound enough.
Comments on the Quality of English LanguageImprovement is required in some places.
Author Response
Comments i: Data in Table 1 needs more clarification. How suddenly did data come?
→ We have provided a description of the data used to calculate probability scores of A-BPs. along with an explanation of the preprocessing steps taken to prepare the dataset for analysis as below. Please refer to the first paragraph in Section 4.1.
The likelihood of A-BPs was determined using data from 2,283 construction accident reports filed with Korea’s Ministry of Land, Infrastructure and Transport’s Construction Safety Management Integrated Information system (CSI), spanning from January 2019 to December 2021 [21]. These reports provided detailed information on accident dates, types of construction, causes, consequences, and incident descriptions. During the data preprocessing stage, the original data required additional categorization, as it did not specify detailed activities or injured body parts. Therefore, keywords associated with reinforced concrete construction (e.g., concrete pouring, formwork, shoring, rebar) were used to create an activity field. Similarly, information on injured body parts was extracted from fields on causes, consequences, and incident descriptions to create an injured body part field. This allowed us to classify accident frequency across 60 A-BPs, representing ten activities and six body parts specific to reinforced concrete construction.
Comments ii: Tables 2 and 3: How is the score authentic per the Author’s opinion? The authors should give the meaning of the data.
→ The scores presented in Tables 4 and 5 are based on a rigorous methodology that utilizes data-driven assessments to ensure the authenticity of the findings. Specifically, probability scores were derived from extensive accident reports filed with Korea’s Ministry of Land, Infrastructure and Transport, covering over 2,000 construction incidents. This dataset provided an objective foundation for calculating the likelihood of injuries associated with specific activity-body part (A-BP) combinations in reinforced concrete construction. The scores reflect actual incident frequencies, classified and averaged to represent realistic risk probabilities across diverse construction activities.
To ensure that the probability scores accurately represent the risk associated with each A-BP, we compared three different methods for score calculation, as detailed in Table 3. The first method involved dividing the frequency range into equal intervals. Although straightforward, this approach resulted in a skewed distribution, where the majority of A-BPs fell into the lowest score category. This imbalance did not accurately capture the variations in injury frequency and was therefore unsuitable for a nuanced risk analysis.
The second method used a ranking-based approach, which aimed to assign scores based on the relative frequency ranks of each A-BP. However, this method encountered practical difficulties due to overlapping frequency ranks, which made it challenging to assign distinct scores consistently. This overlap introduced ambiguity into the scoring system, potentially obscuring the true differences in risk levels between certain A-BPs.
The third method, which we ultimately selected, involved a linguistic classification aligned with occurrence intervals (e.g., "occurring less than once a year," "occurring more than twice a month"). This approach provided a realistic and balanced distribution of probability scores by classifying A-BPs into categories that reflected actual annual and monthly occurrence patterns. This method allowed us to capture a more accurate representation of risk, especially for activities with widely varying frequencies, and facilitated a clearer, more interpretable scoring system for stakeholders.
In conclusion, the third method was chosen for its ability to produce a balanced distribution of scores that aligned closely with real-world injury frequency patterns. By adopting this approach, we ensured that probability scores more effectively represent the relative risk levels across A-BPs, supporting more targeted and practical applications in construction safety management. This selection strengthens the study's foundation for calculating risk scores that can inform tailored safety protocols, personal protective equipment (PPE) guidelines, and worker training programs.
We have added paragraphs to clarify the meaning of the data in Table 5 (previously Table 3 in the initial manuscript). The figures presented in Table 5 provide a detailed perspective on the probability of injury risks across various A-BPs in reinforced concrete construction. Probability scores indicate the likelihood of injuries occurring for specific A-BPs based on historical incident data. Higher scores correspond to A-BPs that have been observed more frequently in accident reports, signaling areas where workers face greater risk due to the nature of the task and the vulnerability of particular body parts.
For instance, a high probability score for wrist-hand injuries during rebar cutting and shaping reflects a higher frequency of incidents in these activities, emphasizing the need for focused safety interventions, such as specialized personal protective equipment (PPE) or ergonomic tools to reduce strain. In contrast, lower probability scores for activities like concrete finishing and cleaning imply a lower observed frequency of injuries in those tasks, suggesting that while safety protocols are still important, these tasks may pose comparatively lower immediate risks to workers.
Comments iii: The introduction has to be more precise and focused. The topics to be covered in subsequent chapters should be clear.
→ We have revised the introduction to improve precision and focus. The primary problem addressed in this study is that activity-based risks and injury characteristics for each body part are often analyzed in isolation, leading to limited discussion on how specific activities impact particular body parts and to what extent. Previous studies have primarily calculated risks associated with individual activities, with minimal research on the relationship between activities and body parts or on risks arising from their combined effects. This issue has been clearly articulated in the revised introduction. Additionally, we have added a paragraph outlining the topics to be covered in subsequent chapters.
Comments v: The manuscript lacks its depth and originality.
→ This study’s originality lies in its calculation of risk scores for risk elements that link specific activities to body parts. To demonstrate the originality of this study, we highlighted its distinction from previous research and its significance in the introduction and literature review sections.
Additionally, we created a new discussion section to provide a more in-depth analysis. The revised discussion section now emphasizes the connections between specific activities and the affected body parts, helping to clarify the nature of these injuries. It includes both practical and theoretical implications. The practical implications offer specific recommendations and actionable measures for safety managers to develop targeted strategies addressing the identified risks on construction sites. These recommendations are closely aligned with our findings to ensure their relevance and applicability. Meanwhile, the theoretical implications position our results within the broader context of construction safety management, establishing a foundation for future research in this field. We believe these additions enhance the manuscript by providing more concrete guidance for developing effective safety strategies.
Comments vi: Conclusion needs to be thoroughly improved or changed.
→ We have expanded the conclusion section to comprehensively summarize the main findings and underscore the significance of this study’s contributions to construction safety management. The conclusion now emphasizes the practical implications of our findings, particularly the use of data on A-BP injury risks to inform customized safety equipment design, tailored training programs, and activity-specific safety protocols. Furthermore, we have outlined directions for future research, suggesting that an expanded study scope and deeper investigation into environmental conditions, seasonal impacts, and other specific factors could provide further insights into risk factors in construction. Please refer to the revised conclusion for these detailed enhancements.
Comments vii: The work is not so sound in the academic study.
→ We respectfully disagree with the assessment that this study lacks academic soundness. This research follows a rigorous methodology designed to quantitatively analyze injury risks in reinforced concrete construction by linking specific activities to body parts, thus filling a notable gap in the construction safety literature. The study systematically builds on existing research by employing a structured risk breakdown that aligns with industry standards, as outlined by organizations such as OSHA and KOSHA. By combining empirical accident data with expert survey responses, we created a robust dataset that allows for statistically valid and practical risk assessment.
This study introduces an innovative Activity-Body Part (A-BP) risk assessment framework, which we believe is both theoretically and practically valuable for construction safety management. While previous studies have primarily examined general risk factors associated with different construction activities, our research goes further by focusing on how specific activities impact distinct body parts. This nuanced approach is intended to provide a clearer understanding of injury risks, which can be directly used to guide more effective, targeted safety interventions on construction sites.
Additionally, we have enriched the discussion section to address the implications of these findings for both research and practice. The practical application of the A-BP framework includes potential improvements in safety protocols, personal protective equipment (PPE) customization, and worker training programs, which are grounded in our data-driven findings. By providing comprehensive guidance for enhancing workplace safety, we hope to demonstrate the study’s contribution to advancing academic understanding and practical application in construction safety management.
Comments viii: The study was based on the conventional construction method. What about the modern construction approach?.
→ We recognize the importance of considering modern construction approaches. This study specifically targeted risks within reinforced concrete construction using conventional methods, as this remains a prevalent practice across numerous construction projects, especially in regions where manual labor and traditional methods are still dominant. Thus, our findings are designed to provide immediately applicable safety insights for the majority of current construction practices worldwide.
However, we acknowledge that modern construction approaches, including prefabrication, modular construction, and increased automation, are becoming more integrated within the industry. These methods inherently involve different safety risks and considerations. For future research, we plan to expand the A-BP framework to address emerging construction techniques, examining how these newer methods impact worker safety differently than conventional approaches.
In summary, while this study provides a focused examination of risks in conventional reinforced concrete construction, we recognize the value of adapting our framework to modern construction methods. Future studies will aim to capture the unique safety profiles of these innovative approaches, offering a broader, adaptable framework that aligns with ongoing advancements in construction practices.
Comments ix: The following topic or its broader area may help you to improve your study.
# Safety barrier warning system for underground construction sites using Internet-of-Things technologies
# Third-party quality control concept, methodology, and implementation in the building and housing construction industry in India
# Accident analysis for construction safety using latent class clustering and artificial neural networks.
# A comparative study of project risk management with risk breakdown structure (RBS): a case of commercial construction in India.
→ Thank you for your helpful suggestions. We appreciate your recommendation to enhance our future research. Several references are directly relevant to our study and have been cited in the main text to provide stronger support for our findings. Including these citations allows us to place our research within the broader context of existing studies, enhancing both the credibility and depth of our analysis.
→ We have improved the quality of the English language throughout our manuscript. These revisions enhance clarity and readability, ensuring that our arguments and findings are presented in a concise and precise manner.
Reviewer 3 Report
Comments and Suggestions for Authors
Abstract: The need for this specific study is not clearly presented.
Introduction:
a. Overall, the introduction contains an insufficient portion of the literature review, leading to a lack of clarity regarding the research gap. It is highly recommended to add additional information to clarify the need for this study and how it differs from existing research.
b. [Line 60-61] Some words are written in Korean.
Background:
a. There are concerns about the writing quality of the background section. The current structure simply lists previous findings, making it difficult to follow the flow of content and understand the relevance of the references. It is highly recommended to completely revise the structure to better justify the research and provide supplementary information.
b. Simply stating the absence of certain assessments without explaining their importance does not justify the gap. Therefore, it would be necessary to add more information regarding the significance of the study.
Methodology:
a. [Table 1] Please check the line for the 5th row.
Results and Discussion:
a. More in-depth analysis is needed to enhance the significance of the study. For example, the linkage between injured body parts and the characteristics of each activity is missing. Further analysis is necessary to understand why these results occur for each activity. This type of deeper analysis could help enhance the study's significance and contribute to the design of a more specific risk management system to prevent injuries.
b. It is suggested that the discussion be written more carefully, with a more comprehensive and deeper interpretation of the findings. Additionally, when referring to literature to support the findings, it is important to reference literature that is highly relevant to the study's specific results. For example, the sentence: "In terms of injured body parts, the overall risk for wrist and hand injuries was high (0.688), with Rebar work (R1-W) recording the highest risk. These results are consistent with Choi (2015), who analyzed that younger workers are particularly vulnerable to finger, hand, and wrist injuries, and similar to Berglund et al. (2019) [11, 13]," is problematic. The dataset in this study was not focused on younger workers, and there was no analysis related to workers' age. Additionally, it is unclear whether the findings from the references are directly related to the reinforced concrete project, which is the focus of this study. Therefore, the references may not be appropriate for supporting the findings.
Comments on the Quality of English LanguageThe paper needs improvement in the overall flow and structure of the content. There are multiple sections that list findings without adequate explanation or connection, which makes the narrative difficult to follow. A clearer organization and better transitions between ideas would enhance the readability.
Author Response
Abstract: The need for this specific study is not clearly presented.
→ We have revised the abstract to present the necessity of this specific study more clearly as below.
Abstract: This study addresses occupational safety in reinforced concrete construction, an area marked by high accident rates and significant worker injury risks. By focusing on activity-body part (A-BP) combinations, this research introduces a novel framework for quantifying injury risks across construction activities. Reinforced concrete construction tasks are categorized into ten specific activities within three major work types: rebar work, formwork, and concrete placement. These are further analyzed concerning six critical body parts frequently injured on-site: head·face, arm·shoulder, wrist·hand, torso, leg·pelvis, and foot·ankle. Using data from 2,283 construction accident reports and expert surveys, the probability and severity of injuries for each A-BP element were calculated. Probability scores were derived from actual incident data, while severity scores were determined via expert evaluations, considering injury impact and required recovery time. To ensure precision and comparability, scores were standardized across scales, enabling a final risk assessment for each A-BP. Results identified that wrist and hand injuries during rebar work activities, particularly cutting and shaping, exhibited the highest risk, underscoring the need for focused protective measures. This study contributes to construction safety management by providing detailed insights into injury risk based on activity-body part interactions, offering safety managers data-driven recommendations for tailored protective equipment, enhanced training, and preventive protocols. This research framework not only helps optimize safety interventions on conventional construction sites but also establishes a basis for future studies aimed at adapting these strategies to evolving construction methods.
Introduction:
a. Overall, the introduction contains an insufficient portion of the literature review, leading to a lack of clarity regarding the research gap. It is highly recommended to add additional information to clarify the need for this study and how it differs from existing research.
→ We have revised the introduction to improve precision and focus. The primary problem addressed in this study is that activity-based risks and injury characteristics for each body part are often analyzed in isolation, leading to limited discussion on how specific activities impact particular body parts and to what extent. Previous studies have primarily calculated risks associated with individual activities, with minimal research on the relationship between activities and body parts or on risks arising from their combined effects. This issue has been clearly articulated in the revised introduction. Additionally, we have added a paragraph outlining the topics to be covered in subsequent chapters.
b. [Line 60-61] Some words are written in Korean.
→ We have corrected the sentence previously written in Korean.
Background:
a. There are concerns about the writing quality of the background section. The current structure simply lists previous findings, making it difficult to follow the flow of content and understand the relevance of the references. It is highly recommended to completely revise the structure to better justify the research and provide supplementary information.
→ We have included Table 1 in the revised manuscript to provide a comprehensive summary of existing research on workers’ injury risks and to clearly illustrate the gap between previous studies and our research. Additionally, we have clearly identified the specific problem this study aims to address in connection with the research gap. Studies on workers’ injury risks typically focused on analyzing the hazards associated with construction activities or assessing the vulnerability of specific body parts within certain construction work. However, these studies often treat activity-based risks and injury characteristics for each body part independently, which limits understanding of how specific construction activities impact certain body parts and to what extent.
In particular, prior research has mainly concentrated on quantifying risks related to individual activities, with limited exploration into the interplay between activities and the specific body parts most affected during these activities. Consequently, there is minimal research addressing the combined effects of activities and body-part injuries, leaving a significant gap in how risk is comprehensively understood and managed in construction settings. Our study addresses this gap by introducing a new approach that calculates risk scores for A-BPs. Instead of isolating activity risk scores, we define these A-BPs as distinct risk elements and calculate risk scores accordingly.
This study specifically addresses the need for improvement in construction safety management by focusing on risks associated with A-BPs within reinforced concrete construction—a sector characterized by high injury rates and labor-intensive work. While prior studies offer a general assessment of risks across construction activities, few have delved into the specific connection between individual tasks and the body parts most susceptible to injury during these activities. By identifying these A-BP-specific risks, our research contributes essential insights that support more tailored and effective safety interventions. This is particularly relevant for reinforced concrete construction, where general safety protocols often fail to capture the detailed risks associated with each activity.
To underscore the importance of our study, we have revised the manuscript to highlight how our framework enables safety managers to develop targeted interventions for each A-BP. This specificity enhances the precision and effectiveness of safety measures by addressing unique risks at the task level. Our approach also facilitates data-driven recommendations that are practical and directly applicable, such as selecting activity-specific personal protective equipment (PPE) and designing focused training that addresses the identified risks in a meaningful way.
In summary, our study not only addresses an underexplored area by analyzing A-BP-specific risks but also provides a valuable foundation for enhancing construction safety practices. By bridging the gap between general activity risks and specific injury risks, our findings offer a more precise and structured approach to construction safety management, which could improve safety protocols across both traditional and modern construction methods. We appreciate the reviewer’s insights, which have guided us in enhancing the clarity and impact of our manuscript to better communicate these significant contributions.
b. Simply stating the absence of certain assessments without explaining their importance does not justify the gap. Therefore, it would be necessary to add more information regarding the significance of the study.
→ We have included Table 1 in the literature review section to provide a comprehensive summary of existing research on workers’ injury risks and to clearly illustrate the gap between previous studies and our research. Additionally, we have clearly identified the specific problem this study aims to address in connection with the research gap. In this study, we address a critical gap in construction safety management by focusing on the specific risks linked to A-BPs within reinforced concrete construction. While prior research has broadly assessed general risks across construction tasks, few studies have focused on the detailed linkage between specific activities and the body parts most vulnerable during these tasks. By identifying these specific risks, our study contributes essential insights that support more targeted and effective safety interventions, especially within labor-intensive reinforced concrete work where injury rates are disproportionately high.
To further highlight the study’s importance, we have revised the manuscript to emphasize that our framework allows safety managers to tailor interventions to each activity-body part, improving the precision and effectiveness of safety measures. This specificity is crucial, as general safety protocols often lack the detail needed to address unique risks posed by each task. Our approach enables data-driven recommendations that are practical and immediately applicable, such as selecting customized personal protective equipment (PPE) and designing focused training that directly addresses identified risks.
In summary, our study not only fills an underexplored gap by analyzing A-BP-specific risks but also provides a practical foundation for improving safety on construction sites. We believe that these targeted insights substantiate the study's significance, as they have the potential to enhance safety protocols across conventional and emerging construction methods. We have updated the manuscript to better convey these aspects and appreciate the reviewer’s guidance in helping us improve its clarity and impact.
Methodology:
a. [Table 1] Please check the line for the 5th row.
→ We have revised Table 1, which is now presented as Table 2 following the revision.
Results and Discussion:
a. More in-depth analysis is needed to enhance the significance of the study. For example, the linkage between injured body parts and the characteristics of each activity is missing. Further analysis is necessary to understand why these results occur for each activity. This type of deeper analysis could help enhance the study's significance and contribute to the design of a more specific risk management system to prevent injuries.
→ Thank you for your constructive comments. We appreciate your suggestion to delve deeper into the linkage between injured body parts and the characteristics of each activity. In response, we created a new discussion section to provide a more in-depth analysis. The revised discussion section now emphasizes the connections between specific activities and the affected body parts, helping to clarify the nature of these injuries. It includes both practical and theoretical implications. The practical implications offer specific recommendations and actionable measures for safety managers to develop targeted strategies addressing the identified risks on construction sites. These recommendations are closely aligned with our findings to ensure their relevance and applicability. Meanwhile, the theoretical implications position our results within the broader context of construction safety management, establishing a foundation for future research in this field. We believe these additions enhance the manuscript by providing more concrete guidance for developing effective safety strategies.
b. It is suggested that the discussion be written more carefully, with a more comprehensive and deeper interpretation of the findings. Additionally, when referring to literature to support the findings, it is important to reference literature that is highly relevant to the study's specific results. For example, the sentence: "In terms of injured body parts, the overall risk for wrist and hand injuries was high (0.688), with Rebar work (R1-W) recording the highest risk. These results are consistent with Choi (2015), who analyzed that younger workers are particularly vulnerable to finger, hand, and wrist injuries, and similar to Berglund et al. (2019) [11, 13]," is problematic. The dataset in this study was not focused on younger workers, and there was no analysis related to workers' age. Additionally, it is unclear whether the findings from the references are directly related to the reinforced concrete project, which is the focus of this study. Therefore, the references may not be appropriate for supporting the findings.
→ We appreciate the reviewer’s feedback on enhancing the depth and clarity of our discussion section. In response, we have revised the discussion to provide a more comprehensive interpretation of the findings. We ensured that the discussion is more closely aligned with the specific scope of our study, particularly focusing on the context of reinforced concrete construction and avoiding any generalizations that may not directly apply to our data. However, when reinforced concrete construction examples were insufficient, we explicitly noted the work type to prevent ambiguity or confusion. This approach allows us to deliver a more focused analysis of the results, ensuring that our interpretations remain relevant to the unique aspects of this study.
We acknowledge the reviewer’s point regarding the references used to support our findings on wrist and hand injury risks, particularly those relating to age. Our dataset did not include age-related variables, and therefore, references that analyze injury risk specifically by age are not directly applicable. In revising this section, we removed references to studies that focus on age-related injury vulnerabilities and instead focused on literature that more closely aligns with reinforced concrete construction activities and their associated risks. In cases where studies on reinforced concrete construction were unavailable, we cited relevant research from other construction types while clearly indicating the context. This adjustment improved the relevance and accuracy of our discussion.
Additionally, we carefully reviewed all references in the discussion to ensure that each cited study directly supports our specific findings within the context of reinforced concrete construction. By prioritizing highly relevant literature, we aim to strengthen the support for our findings and enhance the overall rigor of the discussion. We appreciate the reviewer’s valuable insights, which have helped us identify areas for improvement, and we are confident that these revisions will result in a more focused and substantively grounded discussion.
Comments on the Quality of English Language
The paper needs improvement in the overall flow and structure of the content. There are multiple sections that list findings without adequate explanation or connection, which makes the narrative difficult to follow. A clearer organization and better transitions between ideas would enhance the readability.
→ We extensively revised the introduction, discussion, and conclusion sections. In the introduction, we clearly outlined the focus, significance, scope, methods, and objectives of this study. We also provided an overview of each chapter’s topic at the end of the introduction to guide readers through the structure of the paper.
The previously combined chapter, “Result and Discussion,” was separated into two distinct sections: “Result” and “Discussion.” Findings previously scattered throughout the text were consolidated into the results section, while the discussion section now provides a more in-depth analysis of these findings. The revised discussion emphasizes the connections between specific construction activities and the affected body parts, helping to clarify the nature of these injuries. It also presents both practical and theoretical implications.
Reviewer 4 Report
Comments and Suggestions for Authors
This paper examines the risk of injury to workers' body parts in reinforced concrete construction projects. By classifying work types and body parts, 60 risk elements were identified, and risk scores were calculated based on probability and severity, using accident reports and expert surveys. The findings indicate that formwork and support work pose the highest risks, with wrist and hand injuries being the most frequent. While the study highlights the value of detailed body part classification for risk assessment, it is limited by its reliance on accident reports and the exclusion of fatal accidents. To address some ambiguities in the paper, the following suggestions are made:
1. In the introduction section, the background and significance of the study should be further elaborated to emphasize the critical importance of improving safety management on construction sites. Additionally, the limitations of existing research in addressing these safety concerns should be highlighted to underscore the need for this study.
2. In the introductory section, a brief overview of the specific methodology and steps used to assess risk should be included. This will provide readers with a clearer understanding of the article's structure and content, setting the stage for the detailed discussions in the subsequent sections.
3. In the research methodology section, the specific methodology for quantifying probability and severity should be clearly articulated. This should include details on data sources, data processing methods, and the exact implementation process of the expert survey, ensuring that readers can fully understand the assessment approach employed in the study.
4. In response to the study's findings, specific recommendations and measures should be proposed to guide safety managers in developing targeted safety strategies. These suggestions should focus on mitigating identified risks and enhancing overall safety management on construction sites.
5. In the conclusion section, the main findings and conclusions of the study should be summarized, emphasizing the study's significance and practical applications. Additionally, directions for future research should be proposed, such as expanding the study's scope and conducting in-depth investigations into the impact of specific factors on risk.
6. When explaining how the probability scores were calculated, more details should be provided regarding the three classification methods: Equal Interval Classification, Ranking Method, and Linguistic Classification. Comparisons of these methods should highlight their respective advantages and limitations. Additionally, a thorough explanation of the reasons for selecting Linguistic Classification as the final choice should be included, emphasizing its benefits in the context of the study.
7. Analyze in detail the potential impact of study limitations on the results, such as what biases may result from incomplete and subjective incident reporting. Suggest methods and recommendations to address or mitigate these limitations and how future research could improve them.
8. In the introduction section, when discussing workers' injury risks from various factors (slope disasters), the following literatures are suggested for citation:
[1] Motion characteristics of rockfall by combining field experiments and 3D discontinuous deformation analysis. International Journal of Rock Mechanics and Mining Sciences, 2021, 138, 104591.
[2] Permeability characteristics of fractured rock mass: a case study of the Dongjiahe coal mine. Geomatics, Natural Hazards and Risk, 2020,11(1), 1724-1742.
[3] Discrete element analysis of structural characteristics of stepped reinforced soil retaining wall. Geomatics, Natural Hazards and Risk, 2020, 11(1), 1447-1465.
[4] Field experimental verifications of 3D DDA and its applications to kinematic evolutions of rockfalls. International Journal of Rock Mechanics and Mining Sciences, 2024, 175:105687.
[5] Numerical simulation of wedge failure of rock slopes using three-dimensional discontinuous deformation analysis. Environmental Earth Sciences, 2024, 83: 310.
Comments on the Quality of English LanguageMinor editing of English language required.
Author Response
Comments 1: In the introduction section, the background and significance of the study should be further elaborated to emphasize the critical importance of improving safety management on construction sites. Additionally, the limitations of existing research in addressing these safety concerns should be highlighted to underscore the need for this study.
→ Thank you for your valuable feedback and suggestions. In response, we have revised the introduction to clarify the study’s significance and to highlight the safety concerns associated with activity-body part-based risks. The primary problem addressed in this study is that activity-based risks and injury characteristics for each body part are often analyzed in isolation, leading to limited discussion on how specific activities impact particular body parts and to what extent. Previous studies have primarily calculated risks associated with individual activities, with minimal research on the relationship between activities and body parts or on risks arising from their combined effects. This issue has been clearly articulated in the revised introduction. Additionally, in the literature review, we highlighted the research gap in previous studies to underscore the need for this study and used a table to summarize prior research in order to clarify the distinction between previous research and this study. Please refer to the introduction and literature review sections.
Comments 2: In the introductory section, a brief overview of the specific methodology and steps used to assess risk should be included. This will provide readers with a clearer understanding of the article's structure and content, setting the stage for the detailed discussions in the subsequent sections.
→ We extensively revised the introduction section. In the introduction, we clearly outlined the focus, significance, objectives, scope, methods, and procedure of this study. We also provided an overview of each chapter’s topic at the end of the introduction to guide readers through the structure of the paper.
Comments 3: In the research methodology section, the specific methodology for quantifying probability and severity should be clearly articulated. This should include details on data sources, data processing methods, and the exact implementation process of the expert survey, ensuring that readers can fully understand the assessment approach employed in the study.
→ We have clarified the specific methodology used for quantifying probability and severity. These details, including data sources, data processing methods, and the expert survey implementation process, are articulated in the research methodology section. The probability data were derived from Korea's Ministry of Land, Infrastructure, and Transport’s Construction Safety Management Integrated Information system (CSI) from January 2019 to December 2021, while severity assessments were obtained through an expert survey conducted with 110 industry professionals. Details on the survey design, categorization of activities, and body part-specific injury analysis can be found in the methodology section, with further discussion on interval definitions for probability and severity scores in the Data section. Please refer to the first five paragraphs of Section 3 for a comprehensive outline.
Comments 4: In response to the study's findings, specific recommendations and measures should be proposed to guide safety managers in developing targeted safety strategies. These suggestions should focus on mitigating identified risks and enhancing overall safety management on construction sites.
→ Thank you for your valuable feedback and suggestions. In response, we have expanded the discussion section in the revised manuscript to include both practical and theoretical implications. Please refer to the last two paragraphs of the discussion section. The practical implications provide specific recommendations and actionable measures for safety managers to develop targeted strategies that address the identified risks on construction sites. These recommendations are closely aligned with our study’s findings to ensure that they are relevant and readily applicable.
The theoretical implications, meanwhile, position our results within the broader context of construction safety management, offering a foundation for future research in this field. We believe these additions strengthen the manuscript by providing more concrete guidance for developing effective safety strategies.
Comments 5: In the conclusion section, the main findings and conclusions of the study should be summarized, emphasizing the study's significance and practical applications. Additionally, directions for future research should be proposed, such as expanding the study's scope and conducting in-depth investigations into the impact of specific factors on risk.
→ We have expanded the conclusion section to comprehensively summarize the main findings and underscore the significance of this study’s contributions to construction safety management. The conclusion now emphasizes the practical implications of our findings, particularly the use of data on A-BP injury risks to inform customized safety equipment design, tailored training programs, and activity-specific safety protocols. Furthermore, we have outlined directions for future research, suggesting that an expanded study scope and deeper investigation into environmental conditions, seasonal impacts, and other specific factors could provide further insights into risk factors in construction. Please refer to the revised conclusion for these detailed enhancements.
Comments 6: When explaining how the probability scores were calculated, more details should be provided regarding the three classification methods: Equal Interval Classification, Ranking Method, and Linguistic Classification. Comparisons of these methods should highlight their respective advantages and limitations. Additionally, a thorough explanation of the reasons for selecting Linguistic Classification as the final choice should be included, emphasizing its benefits in the context of the study.
→ To ensure that the probability scores accurately represent the risk associated with each A-BP, we compared three different methods for score calculation. We added Table 3, “Frequency Distribution of Categorization Methods,” to illustrate the frequency distribution of each method and to explain the advantages and disadvantages of each approach. The first method involved dividing the frequency range into equal intervals. Although straightforward, this approach resulted in a skewed distribution, where the majority of A-BPs fell into the lowest score category. This imbalance did not accurately capture the variations in injury frequency and was therefore unsuitable for a nuanced risk analysis.
The second method used a ranking-based approach, which aimed to assign scores based on the relative frequency ranks of each A-BP. However, this method encountered practical difficulties due to overlapping frequency ranks, which made it challenging to assign distinct scores consistently. This overlap introduced ambiguity into the scoring system, potentially obscuring the true differences in risk levels between certain A-BPs.
The third method, which we ultimately selected, involved a linguistic classification aligned with occurrence intervals (e.g., "occurring less than once a year," "occurring more than twice a month"). This approach provided a realistic and balanced distribution of probability scores by classifying A-BPs into categories that reflected actual annual and monthly occurrence patterns. This method allowed us to capture a more accurate representation of risk, especially for activities with widely varying frequencies, and facilitated a clearer, more interpretable scoring system for stakeholders.
In conclusion, the third method was chosen for its ability to produce a balanced distribution of scores that aligned closely with real-world injury frequency patterns. By adopting this approach, we ensured that probability scores more effectively represent the relative risk levels across A-BPs, supporting more targeted and practical applications in construction safety management. This selection strengthens the study's foundation for calculating risk scores that can inform tailored safety protocols, personal protective equipment (PPE) guidelines, and worker training programs.
Comments 7: Analyze in detail the potential impact of study limitations on the results, such as what biases may result from incomplete and subjective incident reporting. Suggest methods and recommendations to address or mitigate these limitations and how future research could improve them.
→ We recognize that the study's reliance on incident reports may introduce limitations related to data completeness and subjectivity, potentially influencing the accuracy of our findings. Below, we discuss the potential impact of these limitations and propose methods to address them in future research.
Comments 8: In the introduction section, when discussing workers' injury risks from various factors (slope disasters), the following literatures are suggested for citation:
[1] Motion characteristics of rockfall by combining field experiments and 3D discontinuous deformation analysis. International Journal of Rock Mechanics and Mining Sciences, 2021, 138, 104591.
[2] Permeability characteristics of fractured rock mass: a case study of the Dongjiahe coal mine. Geomatics, Natural Hazards and Risk, 2020,11(1), 1724-1742.
[3] Discrete element analysis of structural characteristics of stepped reinforced soil retaining wall. Geomatics, Natural Hazards and Risk, 2020, 11(1), 1447-1465.
[4] Field experimental verifications of 3D DDA and its applications to kinematic evolutions of rockfalls. International Journal of Rock Mechanics and Mining Sciences, 2024, 175:105687.
[5] Numerical simulation of wedge failure of rock slopes using three-dimensional discontinuous deformation analysis. Environmental Earth Sciences, 2024, 83: 310.
→ Thank you for your helpful suggestions. This study aims to quantitatively assess injury risks to different body parts of workers during specific activities in reinforced concrete construction through accident data analysis and expert surveys. Among the recommended papers, the first study analyzes the motion characteristics of rockfalls using three-dimensional discontinuous deformation analysis (3D DDA), laying a foundation for disaster prevention and mitigation. While this is an important study, it is not directly relevant to our research focus and was therefore not included. The second study investigates the development and distribution of micro-fractures in the rock mass at the Dongjiahe coal mine, based on 22,517 typical working faces, analyzing the impact of surrounding rock pressure on permeability. Although this is a significant study, we determined it to be outside the scope of our research and thus did not include it.
The third study uses discrete element analysis (DEA) based on a large-scale model test of a stepped Reinforced Soil Retaining Wall (RSRW) to analyze the displacement of particles, stress distribution, force chain distribution, and potential failure surfaces. This research provides valuable insights into the displacement of soil particles and the internal failure mechanisms of RSRW; however, it is not directly related to the injury risks associated with worker body parts in construction, so it was excluded. The fourth study examines the kinetic evolution behaviors of rockfalls through 3D DDA and field experiments near the middle and upper reaches of the Lhasa River in Tibet, with applications to the Wangxia slope in Chongqing, China, evaluating the movement trajectory and changes in kinetic energy of rockfalls quantitatively. Despite its contributions to understanding rockfall dynamics, we determined it to be outside the direct scope of our study.
Lastly, the fifth study explores wedge failure in rock slopes and the subsequent movement and disaster processes using 3D DDA, applying an improved joint contact model to the wedge slope near the Lhasa River in Tibet to analyze the failure and movement of dangerous rock masses. While this research is significant in the context of geotechnical stability, it does not align with our study's focus on injury risks to construction workers, so it was also excluded.
Comments on the Quality of English Language
Minor editing of English language required.
→ We have improved the quality of the English language throughout our manuscript. These revisions enhance clarity and readability, ensuring that our arguments and findings are presented in a concise and precise manner.
Round 2
Reviewer 2 Report
Comments and Suggestions for Authors
Literature review (Comment X), conclusion (Comment vi), question i and ii need improvements.
Author Response
Comments: Literature review (Comment X), conclusion (Comment vi), question i and ii need improvements.
Thank you for your valuable comments. We have further improved the literature review section and addressed questions i, ii, and comment vi. The detailed improvements are explained below. We would greatly appreciate your specific suggestions if any additional improvements are needed.
We have enhanced the literature review section by incorporating Table 1, which provides a comprehensive summary of existing research on workers' injury risks and clearly illustrates the research gap. Specifically, we have added an explanatory paragraph describing Table 1 as follows:
Table 1 provides a systematic review of previous studies in construction safety research, categorizing them based on their analytical focus across three dimensions: Risk of Activity, Injury Risk of Workers' Body Parts, and Risk of Activity-Body Part Combinations. This categorization reveals distinct patterns in research approaches. Studies by Casanovas et al. (2014), Jannadi and Almishari (2003), and Ayhan and Tokdemir (2020) primarily focused on activity-based risk assessment, developing methodologies to evaluate and quantify risks associated with specific construction tasks. In contrast, researchers such as Jeong (1999), Choi (2015), and Berglund et al. (2019) concentrated on analyzing injury patterns related to workers' body parts, examining factors like injury frequency and severity. However, as indicated by the 'X' marks in the last column, none of these studies have attempted to integrate both perspectives by analyzing the specific relationships between construction activities and body part injuries.
Furthermore, we have clearly identified the specific problem this study addresses in relation to the research gap. Our research addresses a critical void in construction safety management by focusing on the specific risks linked to Activity-Body Parts (A-BPs) within reinforced concrete construction. While prior research has broadly assessed general risks across construction tasks, few studies have examined the detailed relationship between specific activities and the body parts most vulnerable during these tasks. By identifying these specific risks, our study provides essential insights that support more targeted and effective safety interventions, particularly within labor-intensive reinforced concrete work where injury rates remain disproportionately high
In response to question i, we have presented the key information of data in Table 2 (previously Table 1): the data source, temporal and spatial scope of the data, and the number of accident cases used in the analysis. Furthermore, we have provided a detailed description of the data used to calculate the probability scores of A-Body Parts (A-BPs), along with the data preprocessing methodology employed for analysis. For comprehensive details, please refer to the fourth paragraph in Section 3. Moreover, we have added introductory sentences describing the data in Table 2 before its detailed explanation.
4th paragraph in Section 3
The probability of A-BPs was determined using data from 2,283 construction accident reports filed with Korea’s Ministry of Land, Infrastructure and Transport’s Construction Safety Management Integrated Information system (CSI), spanning from January 2019 to December 2021 [21]. These reports provided detailed information on accident dates, types of construction, causes, consequences, and incident descriptions. During the data preprocessing stage, the original data required additional categorization, as it did not specify detailed activities or injured body parts. Therefore, keywords associated with reinforced concrete construction (e.g., concrete pouring, formwork, shoring, rebar) were used to create an activity field. Similarly, information on injured body parts was extracted from fields on causes, consequences, and incident descriptions to create an injured body part field. This allowed us to classify accident frequency across 60 A-BPs, representing ten activities and six body parts specific to reinforced concrete construction.
1st paragraph in Section 4
As outlined in the previous section, we organized 2,283 cases of accidents that oc-curred in reinforced concrete construction projects in South Korea from January 2019 to December 2021, categorizing them by activity and the injured body part. Based on this data, we calculated the frequency of injuries to different body parts by activity. Table 2 il-lustrates the distribution of accident-prone body parts (A-BPs) across ten primary con-struction activities. Findings indicate that Form and Support work (F) accounted for the largest proportion of incidents, with 1,635 cases (71.6%), followed by Reinforcement work (R) with 367 cases (16.1%), and Concrete work (C) with 281 cases (12.3%). Notably, the most accident-prone activity was Form positioning and alignment (F3), contributing to 807 cases (35.3%). wrist and hand injuries (F3-W) were found to be the most common within this category, representing 30.5% of the accidents. Conversely, activities such as Concrete finishing and cleaning (C2), along with Concrete curing (C3), were least prone to accidents, with 23 (1.0%) and 28 (1.2%) cases respectively. The analysis also revealed that wrist and hand injuries were the most prevalent, occurring in 779 cases (34.1%), followed by foot and ankle injuries in 324 cases (14.3%). arm and shoulder injuries were compara-tively rare, with only 201 cases (8.8%).
In response to question ii regarding the authenticity and interpretation of scores in Tables 4 and 5(previously Table 2 and 3), we would like to provide a detailed explanation of our methodology and data validation process. The scoring system was developed through a rigorous three-step approach: first, analyzing 2,283 actual accident cases from Korea's Construction Safety Management Integrated Information system (CSI) for probability scores; second, conducting expert surveys with 53 construction safety professionals for severity scores; and third, implementing a standardization process to ensure score compatibility and reliability.
For probability scores (Table 4), we carefully considered three different classification methods before selecting the linguistic classification approach. This method was chosen because it provided the most balanced distribution of scores while maintaining practical interpretability. The frequency ranges were established based on actual occurrence patterns (e.g., 'occurring less than once a year' to 'occurring more than twice a month'), making the scoring system more intuitive and applicable for safety managers. This approach has been validated through comparison with similar methodologies used in previous construction safety research, particularly those focusing on activity-based risk assessment.
Regarding severity scores (Table 5), the data's authenticity is supported by our comprehensive expert survey methodology. The survey participants represented a diverse cross-section of the construction industry, with 82.7% being industry professionals and 15.4% academics, and 36.5% having over 14 years of experience. The severity classifications were aligned with the workplace risk assessment guidelines provided by the Korean Ministry of Employment and Labor, ensuring consistency with established national safety standards. The use of mean scores from expert evaluations helps mitigate individual bias and provides a more reliable assessment of injury severity across different activity-body part combinations.
For Table 4, we now include detailed descriptions of each probability level and its practical implications for safety management. To enhance the meaning and interpretation of the data, we have added a paragraph describing the meaning of the data. In Table 5, we have added descriptive analysis to interpret the meaning of the original probability scores as below. Specifically, we explain that Form and support work shows the highest probability scores, particularly in positioning and alignment activities (F3) with scores of 4-5, indicating frequent accidents. We also highlight that wrist and hand injuries consistently show high probability scores (3-5) across multiple activities, especially in rebar cutting and shaping (R1) and form material preprocessing (F1), while concrete finishing (C2) and curing activities (C3) demonstrate lower probability scores (1-2). This enhanced interpretation of the data provides better understanding of accident patterns and frequencies, helping safety managers identify high-risk activities and prioritize appropriate safety interventions.
Table 5 presents the probability scores for different A-BPs in reinforced concrete construction, derived from the analysis of 2,283 accident cases reported between 2019 and 2021. The probability scores range from 1 to 5, with higher scores indicating more frequent occurrences of accidents. Our analysis reveals notable patterns: Form and support work shows the highest overall probability scores, particularly in positioning and alignment activities (F3) with scores predominantly at 4-5, indicating frequent accidents. Within specific body parts, wrist and hand injuries consistently demonstrate high probability scores (3-5) across multiple activities, especially in rebar cutting and shaping (R1) and form material preprocessing (F1). In contrast, concrete finishing (C2) and curing activities (C3) generally show lower probability scores (1-2), suggesting less frequent accidents.
In response to comment vi, we have revised the conclusion section to improve its quality as follows:
- We have clearly articulated the research contributions, significance, and limitations of this study;
- We have enhanced the conciseness of our sentences while maintaining academic rigor;
- We have strengthened the paragraph flow with more effective transitions; and
- We have eliminated redundant phrases and consolidated related ideas.
7. Conclusion
This study presents a structured approach to assessing injury risks by analyzing specific A-BP combinations within reinforced concrete construction. Through analysis of 2,283 accident reports and expert evaluations, we quantified injury probability and severity for each construction activity. Focusing on distinct activities, this study identifies high-risk A-BPs, particularly highlighting elevated injury risks to wrists and hands during rebar cutting and shaping tasks. These findings are crucial for safety management, providing a detailed risk map for injury-prone activities and enabling effective prioritization of safety interventions. The methodological framework emphasizes the importance of moving beyond generalized risk assessments toward a more nuanced understanding of injury risks in reinforced concrete construction.
Our A-BP classification method emphasizes activity-specific risk elements to enhance construction safety management practices. The findings demonstrate varying risk levels associated with specific activities, supporting the development of targeted protective strategies. This systematic categorization provides comprehensive insights into how construction activities contribute to injury risks, enabling precise risk assessment. The methodology also establishes a foundation for investigating correlations between activity types and injury patterns, facilitating adaptive safety protocols across different construction scenarios.
The practical implications are significant for construction safety management. By providing quantitative A-BP injury risk data, this research enables safety managers to identify vulnerable body parts during specific construction activities. These insights support the customization of safety equipment, worker training programs, and activity-specific safety protocols. For instance, understanding the high-risk nature of wrist and hand injuries in rebar-related tasks can guide the selection of appropriate protective equipment and ergonomic tools. This data-driven approach facilitates proactive decision-making in resource allocation and workplace safety enhancement.
The study acknowledges limitations including potential bias in accident reports due to incomplete or subjective information, and the exclusion of fatal accidents due to data availability constraints. Future research opportunities include expanding the study to other construction types and investigating additional risk factors such as environmental conditions, seasonal effects, and work duration variations. Such expanded analysis could provide more comprehensive insights into construction injury risks, enhancing safety management strategies across diverse construction environments.